# Fracture toughness of a metal–organic framework glass

Theany To [1,4], Søren S. Sørensen[1,4], Malwina Stepniewska [1], Ang Qiao[1], Lars R. Jensen [2], Mathieu Bauchy [3], Yuanzheng Yue [1] & Morten M. Smedskjaer [1✉]

Metal-organic framework glasses feature unique thermal, structural, and chemical properties compared to traditional metallic, organic, and oxide glasses. So far, there is a lack of knowledge of their mechanical properties, especially toughness and strength, owing to the challenge in preparing large bulk glass samples for mechanical testing. However, a recently developed melting method enables fabrication of large bulk glass samples (>25 mm$^3$) from zeolitic imidazolate frameworks. Here, fracture toughness ($K_{Ic}$) of a representative glass, namely ZIF-62 glass ($Zn(C_3H_3N_2)_{1.75}(C_7H_5N_2)_{0.25}$), is measured using single-edge pre-cracked beam method and simulated using reactive molecular dynamics. $K_{Ic}$ is determined to be ~0.1 MPa m$^{0.5}$, which is even lower than that of brittle oxide glasses due to the preferential breakage of the weak coordinative bonds (Zn-N). The glass is found to exhibit an anomalous brittle-to-ductile transition behavior, considering its low fracture surface energy despite similar Poisson's ratio to that of many ductile metallic and organic glasses.

[1] Department of Chemistry and Bioscience, Aalborg University, DK-9220 Aalborg, Denmark. [2] Department of Materials and Production, Aalborg University, DK-9220 Aalborg, Denmark. [3] Department of Civil and Environmental Engineering, University of California, Los Angeles, CA 90095, USA. [4]These authors contributed equally: Theany To, Søren S. Sørensen. ✉email: mos@bio.aau.dk

Metal–organic frameworks (MOFs), which are porous hybrid materials formed from inorganic nodes connected to organic ligands, have potential applications in membrane-based gas sorption, gas separation, shock absorbers, catalysis, and ion transport[1–7]. However, MOF crystals are not available in bulk pieces. The newly emerged family of MOF glasses overcomes this challenge[8,9], and displays numerous interesting features such as relatively high Poisson's ratio[10,11], high transparency[12] and luminescence[13], and in some cases microporosity[14,15]. A certain number of zeolitic imidazolate frameworks (ZIFs)[16–18] can be melted and subsequently quenched to the glassy state before thermal decomposition. ZIFs have topology analogous to those of silica or zeolite networks, whose tetrahedron consists of a metal ion (e.g., $Mg^{2+}$, $Co^{2+}$, $Zn^{2+}$, $Li^+$ and $B^+$) and four imidazolate-based ligands, connected by coordinative bonds[14,16,17,19–21]. An interesting member of the ZIF family is ZIF-62 ($ZnIm_{1.75}bIm_{0.25}$, where Im is imidazolate, $C_3H_3N_2^-$, and bIm is benzimidazolate, $C_7H_5N_2^-$) concerning its ultrahigh glass-forming ability and possibility to be prepared in large size for mechanical tests[10,22]. Structural studies[14,16,17,19–21] have shown that the ZIF-62 glass and crystal exhibit long-range disorder and order, respectively.

Therefore, ZIF-62 glass is in this work chosen to be the object for investigating the mechanical properties of MOF glasses. In contrast to MOF glasses, the mechanical properties of crystalline MOFs have been studied extensively[23]. MOF crystals are typically characterized by relatively low moduli of elasticity[14,23] and hardness[14,23,24], but to our knowledge, the only reported fracture toughness ($K_{Ic}$) values are for two copper phosphonoacetate polymorphs, with $K_{Ic}$ in the range of 0.08–0.33 MPa m$^{0.5}$ as estimated by nanoindentation[25]. More generally, correlations between the mechanical properties and the extent of the network porosity have been observed, showing decreasing moduli and hardness with decreasing density[14,23,26,27]. Given the overall differences in the fracture mechanics of crystalline and glassy materials, such as lack of anisotropy and dislocation-type defects in glasses, there is a need to study the mechanical properties of MOF glasses. To our knowledge, the present work is the first one devoted to understanding the fracture behavior of melt-quenched MOF bulk glasses using a non-indentation technique, being in contrast to two recent studies[11,15]. Li et al.[15] studied the scratch resistance and creep behavior of ZIF-4, ZIF-62, and ZIF-76 glasses by nanoindentation. Using strain-rate jump tests, the ZIF glass members were found to have similar strain-rate sensitivity to glassy polymers and Se-rich chalcogenide glasses. In another indentation study, Stepniewska et al.[11] revealed the anomalous indentation cracking behavior of ZIF-62 glass by atomic force microscopy, and observed apparent shear bands on the indented faces, which are not typically observed in fully polymerized glasses. Although such indentation studies provide useful information on the surface deformation processes, they cannot easily reveal the strength and fracture toughness[28], which are crucial properties for predicting the critical stress and flaw size that induce failure[29]. Indeed, to determine the suitability of MOF glasses for future applications, their mechanical properties including strength and fracture toughness need to be understood. This includes possible application fields within gas storage[30,31], membrane technologies[32], radioactive waste storage[33], and photonics[12,13], which all require knowledge of the underlying mechanical properties to ensure mechanical stability for optimal and reliable performance. However, both measurement and analysis of the fracture toughness of ZIF glasses are challenging for various reasons. First, only relatively small bulk samples of ZIF-62 glass can be produced by melt-quenching (pellets of 11 mm diameter and 1–2 mm thickness). Second, due to its low hardness (~0.6 GPa)[11,15] and low Young's modulus (~3–6 GPa)[11,15], it is difficult to prepare and

process specimens without deformation and breakage. Third, there is currently no standard method to determine fracture toughness of brittle materials shorter than 20 mm in length[34].

Here, we study the fracture behavior of ZIF-62 glass in terms of its fracture toughness and flexural strength. Unlike the existing fracture toughness data for MOF crystals[25], this work reports fracture toughness measurements of a MOF material with a non-indentation method. To overcome the experimental challenges listed above, we have performed careful sample preparation to obtain polished specimens with final dimensions of $1.5 \times 1.9 \times 10$ mm$^3$ and designed a new set-up for single-edge precracked beam (SEPB) measurement of fracture toughness, which complies with the span-beam ratio required in the ASTM standard[34]. To understand the structural origin of the fracture behavior of the ZIF-62 glass, we have also performed molecular dynamics (MD) simulations using a previously validated reactive force field (ReaxFF)[35]. Finally, we have also compared the fracture toughness values to theoretically predicted ones based on the bond strength and the crack path in the glass network. This work contributes to obtaining a general picture about the relation between the fracture and the chemical bonding types for different families of glasses.

## Results

**Molecular dynamics simulations**. We initially validate the ability of the utilized ReaxFF MD potential to replicate the ZIF-62 glass structure. Figure 1a shows the differential correlation function ($D(r)$) from previous X-ray total scattering experiments[27] and the present simulations. Good agreement between the number of peaks and intensities are found, with some minor differences in the peak positions. Overall, this suggests only minor differences between local order structures as determined by experiments and simulation, generally within 0.2 Å and similarly to what has been found for simulated ZIF-4 glass in the original paper introducing the potential[35]. Considering that we also find matching density values (1.60 g cm$^{-3}$ for both present simulations and previous experiments[27]), we use this potential to simulate the fracture mechanism of the ZIF glass, especially given the explicit possibility for bond-breaking to occur in ReaxFF-based MD simulations.

Figure 1b shows an example of a stress($\sigma$)-strain($\varepsilon$) curve starting from a relaxed simulation box that is subjected to tensile strain until fracture to compute the ultimate strength (see Table 1). To estimate the $C_{11}$ and $C_{44}$ components of the stiffness matrix elements for an isotropic material, we calculate the slope of the stress–strain curve in the low-$\varepsilon$ part of a simulation with smaller strain steps, as described in the Methods section (see inset of Fig. 1b). We show similar plots of the other used structures in Supplementary Fig. 1. The determined stiffness constants are then used to calculate the Young's modulus ($E$) and Poisson's ratio ($\nu$), which are found to be $4.1 \pm 0.9$ GPa and $0.395 \pm 0.035$, respectively. The $E$ value obtained from the simulation is slightly lower than that obtained from ultrasonic echography (Table 1) and slightly higher than that previously measured by Brillouin spectroscopy[10]. The $\nu$ value obtained from the simulation is also in between the values obtained from ultrasonic echography (Table 1 and ref. [11]) and Brillouin spectroscopy[10].

Next, we consider the simulated fracture behavior of the ZIF-62 glass. To this end, we employ the in silico method of Brochard et al.[36], which has successfully been used to estimate the fracture toughness of a number of materials, including $\alpha$-cristabolite[36], silica glass[37], calcium aluminosilicate glass[37], and calcium-silicate-hydrate gels[38]. The method involves the introduction of a precrack in the sample (to induce stress concentration), which is subsequently elongated normal to the longest dimension of the

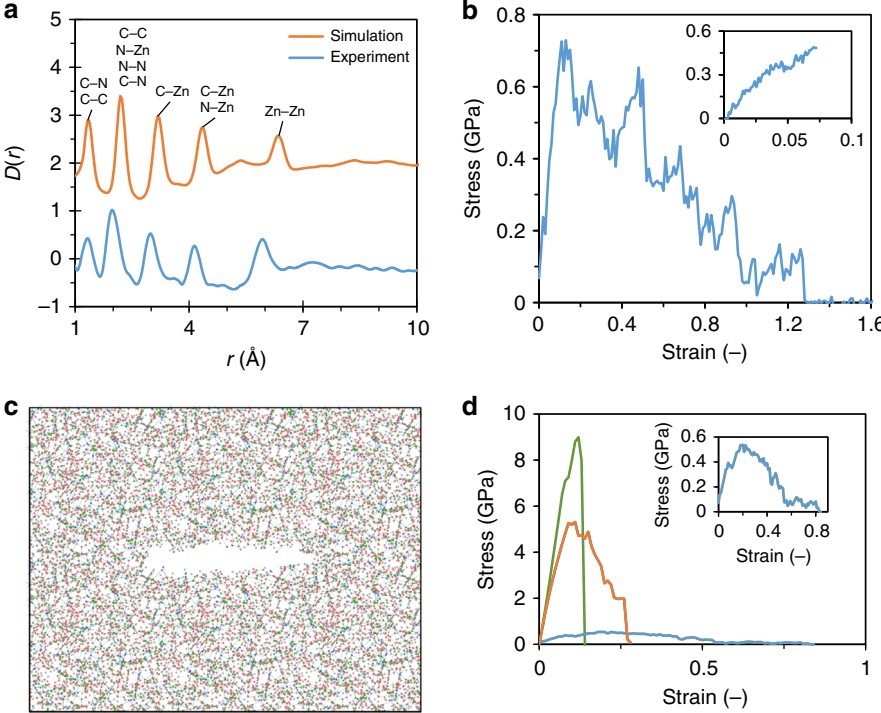

**Fig. 1 Molecular dynamics simulations of ZIF-62 glass structure and mechanical properties. a** Comparison between experimental[27] (blue) and simulated (orange) differential correlation function ($D(r)$) of ZIF-62 glass averaged over eight performed quenches. Data have been shifted vertically by +2 for easier comparison. **b** Example of a stress ($\sigma$) vs. strain ($\varepsilon$) curve used in the estimation of the simulated ultimate strength (main figure) and modulus (inset). **c** Structural representation of an induced precrack in the glass network. Colored spheres represent carbon (red), hydrogen (gray), nitrogen (green), and zinc (blue). Cutoffs for bonds are 2.0 Å for C–C and C–N bonds, 3.0 Å for Zn–N bonds, and 1.5 Å for C–H bonds. **d** Simulated stress–strain curve of the precracked ZIF-62 glass (blue line, enlarged in inset) used in the estimation of fracture toughness. The green and orange lines represent stress–strain curves for disordered $a$-SiO$_2$[37] and calcium aluminosilicate[37], respectively. Note that these simulations featured a slightly different crack-to-box ratio (~0.33) compared to that in this study (~0.41). Source data are provided as a Source Data file.

**Table 1 Comparison of mechanical properties of experimental and simulated ZIF-62 glasses.**

|  | $E$ (GPa) | $\nu$ (−) | $K_{Ic}$ (MPa m$^{0.5}$) | $\gamma$ (J m$^{-2}$) | $\sigma_{max}$ (GPa) | $\varepsilon_{\sigma_{max}}$ (%) |
|---|---|---|---|---|---|---|
| ZIF-62 glass (exp) | 5.2 | 0.343 | 0.104 | 0.82 | 0.008 | 0.3 |
| ZIF-62 glass (MD) | 4.1 | 0.395 | 0.097 | 0.98 | 0.703 | 17.38 |

Estimated errors for elastic modulus ($E$), Poisson's ratio ($\nu$), fracture toughness ($K_{Ic}$), fracture surface energy ($\gamma$), strength ($\sigma_{max}$), and strain at maximum stress ($\varepsilon_{\sigma_{max}}$) are 0.9 GPa, 0.035, 0.009 MPa m$^{0.5}$, 0.11 J m$^{-2}$, 0.087 GPa, and 6.7%, respectively, for MD simulations and 0.3 GPa, 0.001, 0.019 MPa m$^{0.5}$, 0.31 J m$^{-2}$, 0.002 GPa, and 0.07%, respectively, for experiments. Source data are provided as a Source Data file.

crack to induce bond rupture (i.e., mode I fracture). By computing the stress related to the forced elongation in the direction of strain, we obtain a stress–strain curve of the fracture process, which in turn can be used to estimate the fracture surface energy ($\gamma$). Figure 1c shows an example of an unrelaxed induced crack, while Supplementary Fig. 2 shows the same crack after relaxation, revealing notable relaxation of the structure around the crack, yet with some reformation of bonds across the crack surface. Similar or even less bond reformation has been found for other quenches. We ascribe this large relaxation of the crack to the ductile nature of the system at the nanoscale, but it has a negligible influence on the results of the following fracture simulations, as only 0.6 ± 0.2% of bonds are reformed during the structural relaxation of the crack.

Figure 1d presents an example of a stress–strain curve for the precracked ZIF-62 glass. For comparison, we included two other amorphous materials, $a$-SiO$_2$[37] and a calcium aluminosilicate (CAS) glass[37], for which the fracture process has been simulated using the same procedure with an induced precrack, although

with a slightly different crack-to-box ratio. In the inset of Fig. 1d, the stress–strain curve of glassy ZIF-62 is magnified to highlight the large degree of nanoductility even in the presence of a precrack. While $a$-SiO$_2$ features a very brittle fracture behavior, both the CAS glass and ZIF-62 glass show evidence of nanoductility. In the latter, complete fracture is typically achieved at an extreme strain of 70–130%. The remaining stress–strain curves are shown in Supplementary Fig. 3. Based on these data, we compute the critical energy release rate ($G_C$) and $K_{Ic}$ to be 1.96 ± 0.22 J m$^{-2}$ and 0.097 ± 0.009 MPa m$^{0.5}$, respectively. For one simulated structure, we increased the crack size (from 36 Å to 42 and 48 Å) and found no apparent relation between crack size and the resulting fracture toughness (see Supplementary Fig. 4). As it will be shown below, these results are in excellent agreement with the present experiments. For comparison, the values are roughly an order of magnitude lower than those for silica glass and a factor of 3-4 lower than those for the CAS glass[37].

To quantify the degree of ductility, we evaluate the so-called brittleness index[38] as $B_{index} = 2\gamma_s G_C^{-1}$, where $\gamma_s$ is the surface

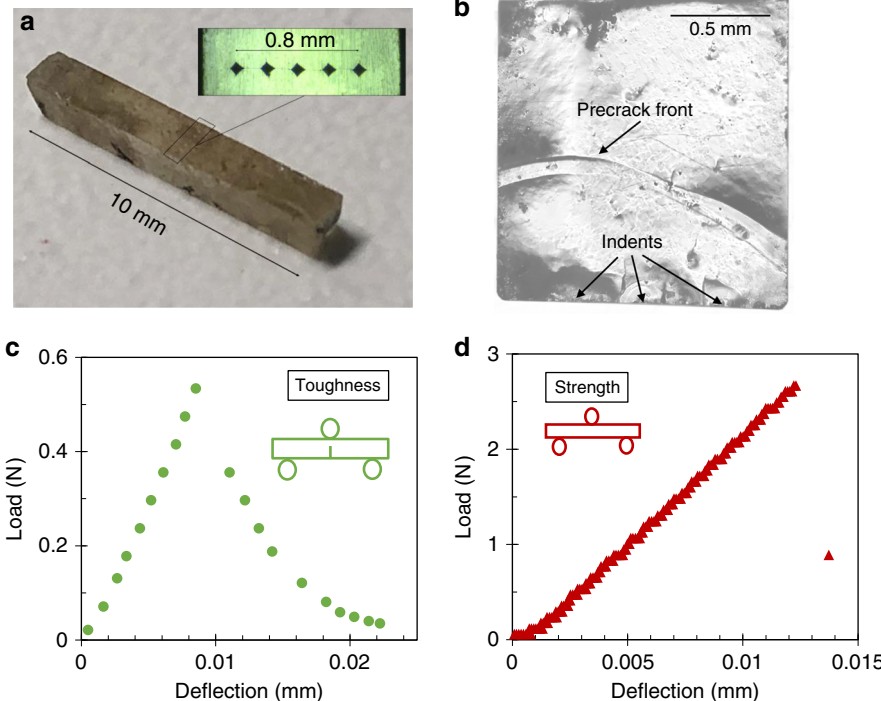

**Fig. 2 Fracture toughness and strength measurement of ZIF-62 glass. a** Example of an indented single-edge precracked beam (SEPB) specimen of ZIF-62 glass with dimensions of 1.5 × 1.9 × 10 mm³. The indentation line is enlarged and shown in the inset. **b** Post-fractured SEPB specimen. **c** Example of load-deflection curve of three-point bending on SEPB specimen. **d** Example of load-deflection curve of three-point bending on non-cracked specimen. Source data are provided as a Source Data file.

energy estimated in a separate simulation (see Methods for details). We estimate the surface energy to be 0.69 J m$^{-2}$, in good agreement with that reported previously for crystalline ZIF-8 (0.43–0.72 J m$^{-2}$)[39]. This brittleness parameter thus gives a measure for how brittle or ductile a fracture is, where 1 is indicative of a perfectly brittle material (i.e., for which the fracture energy is solely used to create some new surface). We find the ZIF-62 glass to have $B_{index} = 0.70$, which is fairly similar to the value obtained for a disordered calcium-silicate-hydrate gel (0.62)[38], but well below that of $a$-SiO$_2$ glass (0.90). Despite the presence of nanoductility, the ZIF-62 glass shows no evidence of macro-ductility as it will be discussed in the following.

**Fracture toughness and strength measurements**. Figure 2a–c present the single-edge precracked beam (SEPB) method for determining $K_{Ic}$ of glassy ZIF-62. SEPB is known as a self-consistent fracture toughness method[29,34] for both ceramics and glasses. However, currently, the longest possible sample dimension of the ZIF-62 glass (10 mm) is lower than the shortest one recommended in the ASTM standard (20 mm)[34]. However, the three-point bending span of 7.5 mm (four times the beam height of 1.9 mm) is in the norm of the span-beam ratio of the ASTM standard[34]. To validate the present adapted SEPB method (see Methods section), we have initially used it to measure $K_{Ic}$ of a standard soda lime silicate glass sample with dimensions of 1.5 × 1.9 × 10 mm³. The obtained result (0.72 ± 0.02 MPa m$^{0.5}$, see Supplementary Table 1) is in excellent agreement with that obtained using the standard SEPB method (0.70 ± 0.01 MPa m$^{0.5}$)[29,40].

Figure 2a shows a typical Vickers-indented beam specimen, including an enlargement of the indentation area to show the typical five indents using a load of 2.5 N on the broadest side ($B$ = 1.5 mm) of the beam. The five indents remain visible in the post-fracture image of the SEPB specimen, i.e., the precrack starts from the indents (Fig. 2b). The uneven state of the precrack front is not

significant. By measuring the precrack length at various fractions (25%, 50%, and 75%) of $B$, we obtain the difference between the average precrack length and any measurement at 0.25$B$, 0.50$B$, and 0.75$B$ to be acceptable according to the ASTM standard (less than 10%)[34]. An example of the load-deflection curve from the three-point bending experiment on the precracked specimen is shown in Fig. 2c, revealing a partially unstable fracture with instability at the maximum load ($P_{max}$), which is required for not being influenced by stress corrosion[40]. $K_{Ic}$ is then calculated from the precrack length and $P_{max}$ (see Methods section).

Table 1 presents a comparison of the results from SEPB experiments and MD simulations. Experimental $K_{Ic}$ (0.104 ± 0.02 MPa m$^{0.5}$) and $\gamma$ (0.82 ± 0.31 J m$^{-2}$, calculated using Eq. (8) in Methods) values agree well with the simulated ones, thus revealing low damage tolerance of the hybrid organic-inorganic network glass. The three-point bending strength is determined to be 8 ± 2 MPa (Fig. 2d), which is two orders of magnitudes lower than the simulated value (703 ± 87 MPa). Accordingly, the corresponding strain at fracture for the experiment is two orders of magnitude lower than that for the simulation. We infer that this difference is due to the difference in the surface flaw size distribution in the experimental specimen and the ideally flawless and surfaceless simulated specimen, as well as the major difference in sample size between experimental and simulated specimens. As a brittle solid, the surface flaw dependence of strength of experimental ZIF-62 glass is significant[41]. Similarly, the size of the simulated specimen (nanometer scale) limits the size of the obtainable defects, thus lowering the possibility of having defects with critical dimensions. In other words, experimental strength varies primarily with the surface flaw size and not with the strength of the flawless bulk material. This is also the reason why we introduce a precrack in both simulation and experiment to induce stress concentration and remove effects of surface flaws on fracture behavior.

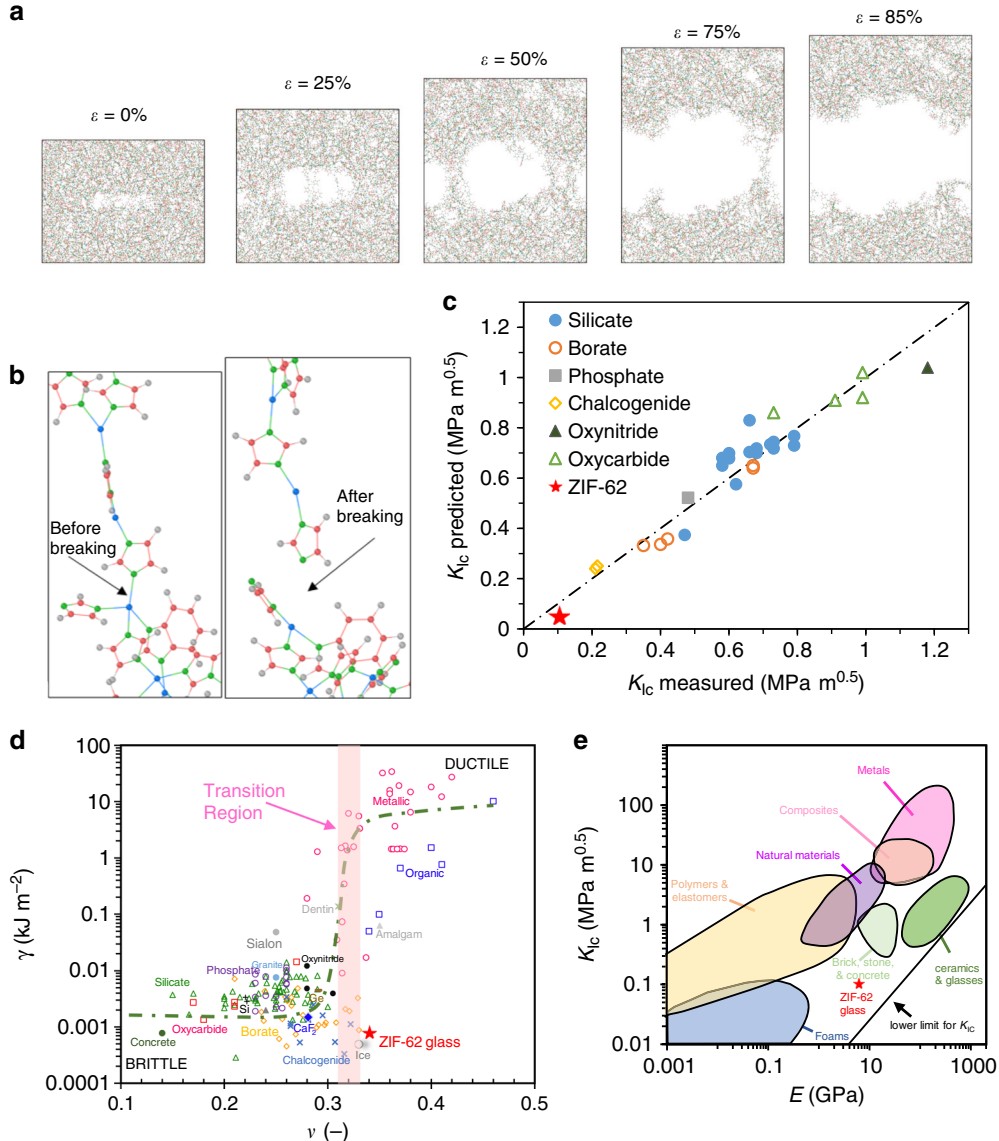

**Fig. 3 Fracture mechanism of ZIF-62 glass and comparison with other material families. a** Structural representation of the crack propagation in the precracked ZIF-62 glass upon increasing strain ($\varepsilon$). Colored spheres represent carbon (red), hydrogen (gray), nitrogen (green), and zinc (blue). **b** Enlarged view of the Zn–N bond before (left) and after breaking (right). **c** Comparison of theoretically predicted and experimental fracture toughness ($K_{Ic}$) for a range of glass and glass-ceramic materials. The theoretical prediction is explained in the text. Figure is adopted with data from ref. [44], in addition to data for silicate and borate glasses[54], oxycarbide glass-ceramics[45], and the present ZIF-62 glass. All the experimental $K_{Ic}$ values are from self-consistent methods such as SEPB, chevron notched beam, and surface cracked in flexure, with an error smaller than ±0.05 MPa m$^{0.5}$. **d** Relationship between fracture surface energy ($\gamma$) and Poisson's ratio ($\nu$) for a range of materials. Figure is adopted with the data from refs. [48,49,62] and extended with additional data for metallic glasses[63–66], silicate glasses[67,68], borate glasses[69–71], chalcogenide glasses[70,72,73], phosphate glasses[70,74], fluoride glasses[68,70], oxycarbide glasses and glass ceramics[45], tellurite glass[70], and the present ZIF-62 glass. **e** Ashby plot of the relation between $K_{Ic}$ and Young's modulus ($E$) for a range of materials. The figure is adopted with data from ref. [75] and extended with that of the present ZIF-62 glass. Source data are provided as a Source Data file.

**Structural origin of ZIF-62 fracture behavior**. To understand the structural origin of the low fracture toughness and fracture energy of the ZIF-62 glass observed in both experiments and simulations, we investigate the atomic-scale crack propagation mechanism using the ReaxFF-based MD simulations. By considering a representative example of the atomic configuration of the glass with an induced precrack, we observe an initial stretching of bonds both within the crack and in the bulk areas at the ends of the crack for small strain values (Fig. 3a). Upon increasing strain, actual bond breaking initiates and the crack then propagates by increasing its width. Interestingly, nanocavities also appear, found to be in the same plane as the precrack, while smaller voids are observed within the bulk glassy phase. The

appearance of cracks not directly in conjunction with the pre-crack gives rise to molecular bridges across the two fracture surfaces (Fig. 3a), causing the observed nanoscale ductility in the stress–strain curve (Fig. 1d).

This fracture behavior is largely similar to previous experimental observations of crack propagation by initiation, growth, and coalescence of nanocavities[42]. Noticeably, such mechanism has been argued to provide nanoductility while maintaining brittleness at the macroscale, in agreement with the findings herein. The molecular bridges consist of two- to four-fold coordinated Zn atoms. In many cases, the bond breaking occurs for the two- and three-fold coordinated Zn atoms, but interestingly, visual inspection reveals that when four-fold coordinated sites are present, they

also appear to break instead of nearby two- and three-fold coordinated Zn atoms under intensive strain (Fig. 3b). As such, it is the relatively weak Zn–N coordinative bonds in the present ZIF-62 glass compared to the stronger bonds in other categories of glasses (e.g., oxide or metallic) that ultimately gives rise to its low fracture toughness. This argument is based on an analysis of the simulated bond breaking during fracture, in which we find that only Zn–N bonds are breaking. However, surprisingly we found a larger number of total bonds *after* full fracture compared to before the forced deformation of the simulation box (see Supplementary Fig. 5). This is caused by structural rearrangement stimulated by the applied stress, ultimately increasing the connectivity of the network. In addition, we performed a stress analysis during fracture on all eight simulated structures, illustrating that Zn, N, and C atoms all carry stress during fracture, as expected since they are the backbone of the network (see Supplementary Figs. 6 and 7). These findings, combined with the relatively low bond energy of Zn–N[43], explains why only these bonds break during fracture.

To quantitatively understand the low fracture toughness and verify the assumption about the importance of the coordinative bonds as found by MD simulations, we next compare the experimental $K_{Ic}$ with a theoretical one obtained using Rouxel's recent model[44] (Fig. 3c). In this approach, fracture toughness is predicted by means of the similarity principle, $K_{Ic}^{the.} = \sqrt{2\gamma^t E'}$ (here $E' = E/(1 - v^2)$ for plane strain), where $E$ and $v$ are here taken as the experimental values from ultrasonic echography and $\gamma^t$ is the theoretical fracture surface energy. In turn, $\gamma^t$ is predicted based on the experimental density ($\rho$), the molar mass of the glass, the interatomic bond strength, and the bond concentration along a fracture surface[44]. Here, we first assume that the network topology of ZIF-62 glass is analogous to that of amorphous silica. The central atom in ZIF-62 glass is Zn (Si in $a$-SiO$_2$), which is connected to three Im ligands and one bIm ligand (four atoms of O in $a$-SiO$_2$). For the crack to propagate, we further assume that the crack breaks one Zn–N bond in the ZnX$_4$ unit (where X is Im or bIm) before proceeding to the next ZnX$_4$ unit. We assume a bond energy of 160 kJ mol$^{-1}$, which is the value found for the Zn–N bond in ZIF-zni[43], a crystal having similar composition to ZIF-4, yet with a denser crystal structure and thus a good representative of the bonding found in the glassy state of ZIF-4 and ZIF-62. By applying the model, we calculate $K_{Ic}$ of ZIF-62 glass to be 0.06 MPa m$^{0.5}$ (Fig. 3c), in reasonable agreement with both the experimental and MD-simulated $K_{Ic}$ values (Table 1). The slight underestimation may be explained by the difference between O in SiO$_2$ and X in ZnX$_2$, since X here is not just an atom, but an organic molecule (i.e., a large 3D-dispersed structure) in the glass network. That is, the crack should need more energy to (i) escape from the strong intramolecular bonds of Im and bIm in the ZIF-62 glass and (ii) form the nanocavities in tensile stress before fracture as explained above. In fact, the variation in energy between Rouxel's theoretical model and that obtained from experiments (or MD simulations) is equal to the difference of half the critical energy release rate and the surface energy, i.e., $G_C/2$ - $\gamma_s$, suggesting that the theoretical model used to calculate $\gamma^t$ covers only the elastic part of the system. On the other hand, following ref. [45], we can predict the needed energy by assuming that the energy required to escape from the strong intramolecular bonds of Im and bIm is equal to that required to break the weakest bonds of bIm, as bIm is larger in volume than Im. By applying this approach, we found a very good agreement of $K_{Ic}$ between model prediction (0.097 MPa m$^{0.5}$) and experiment (0.104 MPa m$^{0.5}$). As an alternative view, we may estimate the required bond energy directly from the experimental $K_{Ic}$ value under the similar set of assumptions. In this case, we obtain a bond energy of 471 kJ mol$^{-1}$, which is considerably higher than

the bond energy of Zn–N (160 kJ mol$^{-1}$)[43], yet lower than those of C=C (>600 kJ mol$^{-1}$) and C≡N (>750 kJ mol$^{-1}$). This again stresses that the observed fracture requires considerably more energy than a purely brittle one, highlighting the need to consider cohesive and plastic effects when estimating $K_{Ic}$.

Finally, we note that the present MOF glass exhibits an anomalous behavior in the fracture surface energy-Poisson's ratio ($\gamma$-$v$) landscape (Fig. 3d), in which a sharp brittle-to-ductile transition exists at Poisson's ratio value around 0.32 for various non-crystalline materials[46–49]. That is, although the present MOF glass features a relatively high Poisson's ratio in the range of many metallic and organic glasses, it has a much lower fracture surface energy, which is similar to that of the lower Poisson's ratio materials like ice, chalcogenide glasses, and some borate glasses. Moreover, the MOF glass does not belong to any other group of materials in the Ashby plot of $K_{Ic}$ vs. $E$ (Fig. 3e). Although ZIF-62 glass has a Young's modulus similar to that of the group of natural materials, it exhibits lower fracture toughness in the range between foams and elastomers.

## Discussion

The modified SEPB experiment for small (10 mm in length) samples, which has been validated on standard window glass, has enabled us to measure the fracture toughness of the ZIF-62 glass. The validity of the small SEPB setup to measure $K_{Ic}$ is two-fold. First, the span-to-width ($S/W = 7.5/1.9$) ratio of the new setup is about 4, as permitted by ASTM standard ($4 \leq S/W \leq 10$)[34] and the width-to-broadness ($W/B = 1.9/1.5$) ratio is about 1.27, as also permitted by the standard ($1 \leq W/B \leq 2$). Second, the use of an indentation line to join the corner cracks lowers the compression load in the precracking stage and thus facilitates the straight through precrack, i.e., the low compression load ensures that the indented specimen produces a precrack before compressive breaking. We expect that this modified SEPB setup can also be applied to other families of glasses and ceramics, which can only be produced in small size and quantity as the present MOF glass.

Based on the ReaxFF-based MD simulations, we have found that the rupture of weak Zn–N coordinative bonds is the origin of the low fracture toughness of the ZIF-62 glass. Taken as a whole, this glass features a number of unique properties, such as pressure-promoted melting[50], ultra-high resistance to crystallization[10], and broad mid-infrared luminescence[13], and, as shown herein, also uniqueness in its fracture behavior. First, it features pronounced nanoductility compared to other glasses e.g. silica[37] (Fig. 1d), but yet it exhibits very low fracture toughness on the macroscale, similarly to that of elastomers or high-toughness foams (Fig. 3e). Moreover, with this low toughness, the ZIF-62 glass possesses a much larger Young's modulus ($E = 4$–6 GPa) than foams and elastomers ($E < 1$ GPa). These characteristics position the MOF glass in a hitherto unoccupied region of the Ashby $K_{Ic}$ vs. $E$ plot (Fig. 3e). The small fracture toughness and relatively large Young's modulus gives rise to a small fracture surface energy ($\gamma$), comparable to that of oxide glasses. With a Poisson's ratio ($v = 0.34$–0.35) as high as many metallic and organic glasses, the studied hybrid glass shows another anomalous feature in the brittle-to-ductile transition curve (Fig. 3d). This work thus has a far-reaching implication for understanding the fracture behavior and mechanical properties of the entire family of MOF glasses.

## Methods
**Experimental sample preparation**. A metal–organic framework glass, namely ZIF-62 glass, was produced by melt-quenching crystalline ZIF-62 in a tube furnace under flowing nitrogen gas. The solvothermal synthesis of the crystalline ZIF-62 was previously reported in details[10–12,51,52]. Briefly, the mixture of zinc nitrate

hexahydrate $(Zn(NO_3)_2)\cdot6H_2O$, Merck), imidazole $(C_3H_4N_2$, Merck), and benzi-midazole $(C_7H_6N_2$, Merck) solutions in $N,N'$-dimethylformamide (DMF, VWR) was magnetically stirred for 20 min, before placing it in a furnace for 48 h at 403 K. The obtained crystalline powder was washed thrice in DMF, once in dichlor-omethane (DCM, Merck), and then dried in a furnace at 373 K for 4 h. To confirm the obtained powder was ZIF-62, powder X-ray diffraction was performed (see Supplementary Fig. 8). The crystalline powder (200 mg) was pressed uniaxially (pressure of 40 MPa) into a pellet with a diameter of 13 mm and a thickness of ~2 mm. The pellet was heated to 738 K with a heating rate of 10 K $min^{-1}$ and held isothermally for 5 min in an electrically heated tube furnace with the nitrogen gas flow set to 5 mL $min^{-1}$. Then the sample was cooled down to room temperature (~295 K) at a cooling rate of 10 K $min^{-1}$. The final ZIF-62 glass is a bulk homo-geneous glass with 11 mm diameter without visible bubbles[52].

The 11 mm diameter glass pellet was carefully cut into beams of dimension of $2 \times 3 \times 10$ $mm^3$ by using a thin diamond disk (thickness of 0.15 mm). To cut the cracking-prone ZIF-62 glass without breaking, we first tested cutting under various rotation speeds of the disk and translation speeds of the specimen. We finally chose the rotation and translation speeds of 2000 rpm and 0.015 mm $s^{-1}$, respectively. To grind and polish the ZIF-62 glass without breaking, we mounted it together with three pieces of standard window (soda-lime-silicate) glass with similar dimensions. This was done to the four edges of the specimen to have the final dimension of about $1.5 \times 1.9 \times 10$ $mm^3$ after grinding and polishing down to 9 μm diamond paste.

**Experimental elastic properties and strength.** The Young's modulus $(E)$ and Poisson ratio $(v)$ of the prepared ZIF-62 glass were measured by means of ultra-sonic echography. Generated by 10 MHz piezoelectric transducers, the velocities of longitudinal $(V_L)$ and transverse $(V_T)$ waves were calculated. Using the density $(\rho)$ measured by means of Archimedes' method in distilled water at room temperature (295.8 K), $E$ and $v$ were then obtained by means of the following relations,

$$E = \rho \frac{3V_L^2 - 4V_T^2}{\left(\frac{V_L}{V_T}\right)^2 - 1} \qquad (1)$$

$$v = \frac{E}{2\rho V_T^2} - 1. \qquad (2)$$

To obtain the flexural strength of the prepared glass, the as-polished specimen was mounted on a three-point flexure with a supporting span of 7.5 mm between two rollers with diameter of 4 mm. The mounted specimen was loaded with a cross-head speed of around 0.06 mm $min^{-1}$, depending on the specimen dimension, to have a strain rate of $1 \times 10^{-4}$ $s^{-1}$ according to the ASTM standard C1161-13[53] at room temperature (20 °C) and relative humidity of 60%. The average strength was calculated from four tests.

**Experimental fracture toughness.** Fracture toughness $(K_{Ic})$ was determined by means of the single-edge precrack beam (SEPB) method adapted for small dimension specimen $(1.5 \times 1.9 \times 10$ $mm^3)$, following the experimental procedure given in ASTM standard C1421-10[34] and recent literature[40,45,54]. In this modified SEPB approach, five Vickers indents with a load of 2.5 N for a dwell time of 5 s were placed on a line (200 μm from one indent to another) on the broadest side $(B = 1.5$ mm), as shown in Fig. 2a. The indented specimen was positioned in the compression fixture with a groove size of about 3 mm (~1.5 times the specimen width, $W = 1.9$ mm) to produce a precrack (before the fracturing step) with a cross-head speed of 0.02 mm $min^{-1}$. Under the compression fixture, the specimen experiences tensile stress in the lower part (indented or groove part) and com-pressive stress in the upper part. The tensile stress opens up the crack from the five indents, and the crack grows until it reaches the compressive stress (about the middle of the specimen width). This prevents further extension of the precrack and we obtain a precrack with the size about half-length of $W$. A three-point bending fixture, as the one used in the flexural strength testing, was used to fracture the precracked specimen with a cross-head speed of 10 μm $s^{-1}$ to avoid the humidity effects (see ref. [40] for details). We note that this adapted three-point bending span $(S)$ of 7.5 mm was designed to fulfill the span-to-width ratio of about 4 as required in the standard[34]. $K_{Ic}$ was then calculated from the peak load $(P_{max})$[34,40,55],

$$K_{Ic} = \frac{P_{max}}{B\sqrt{W}} Y^* \text{ where } Y^* = \frac{3}{2} \frac{S}{W} \frac{\alpha^{1/2}}{(1-\alpha)^{3/2}} f(\alpha), \qquad (3)$$

where $\alpha$ is the precrack-width ratio $(a/W)$ and $f(\alpha) = [1.99 - (\alpha - \alpha^2)(2.15 - 3.93\alpha + 2.7\alpha^2)]/(1 + 2\alpha)$. We validated this adapted SEPB approach for small specimens on standard window glass (soda-lime-silicate glass), giving the value $K_{Ic} = 0.72 \pm 0.02$ MPa $m^{0.5}$, which agrees with that obtained from SEPB experiments on larger specimens[40]. The average $K_{Ic}$ value is calculated from five valid tests.

**MD simulations of glass formation.** All simulations were performed using LAMMPS[56]. As a starting point, we used the unit cell structure of crystalline ZIF-62 from the work of Widmer et al.[50], which was then replicated into a $2 \times 2 \times 2$ supercell (2368 atoms) for better statistical averaging of structures. To prepare the ZIF-62 glass, we employed the ReaxFF parameters as introduced by Yang et al.[35] since the ReaxFF bond order potential is ideally suited for the bond breaking and

reformation that are needed to study the melting and fracture characteristics of the MOF glass herein[35,57]. To prepare the glassy structure, we performed potential energy minimization of the initial unit cell structure, followed by 7.5 ps of relaxation at 10 K using a Berendsen thermostat. The structure was then heated to 300 K at a rate of 232 K $ps^{-1}$ with 12.5 ps of equilibration at 300 K. Afterwards the structure was heated to 1500 K at 24 K $ps^{-1}$ and again cooled to 300 K at a similar rate before equilibration of the structure for another 12.5 ps, and finally 6.25 ps of statistical averaging. All quenching steps, besides the final statistical averaging, were performed in the $NPT$ ensemble at 1 atm, while the statistical averaging was per-formed in the $NVT$ ensemble. Eight structures were made by varying the initial temperature profile induced when starting dynamics. A timestep of 0.25 fs and periodic boundary conditions in all three directions were used for all steps of MD simulations including all following analyses. We have provided an example structure of the ZIF-62 glass as a Supplementary File in the standard LAMMPS data file format, referring to the LAMMPS documentation for an introduction to the data format and visualization software for format conversion (e.g., OVITO).

**MD simulations of elastic properties and ultimate strength.** The obtained melt-quenched glass structures were subjected to elastic property analysis at 300 K by stepwise deforming the relaxed simulation box in strain steps of $\varepsilon \approx 0.001$ for elongation and $\varepsilon \approx 0.003$ for shear deformation in the $NVT$ ensemble. After each step, the structure was first equilibrated for 2.5 ps, while the next 2.5 ps of simu-lation time was used for averaging the pressure in the elongated direction. The linear parts of the obtained stress–strain curves were used to obtain the stiffness matrix elements $C_{11}, C_{22}, C_{33}, C_{44}, C_{55},$ and $C_{66}$. Under the assumption of an isotropic nature of the glass structure, we calculated the $C_{11}$ as an average of $C_{11}$, $C_{22}$, and $C_{33}$ and $C_{44}$ as an average of $C_{44}, C_{55}$, and $C_{66}$. Next, we calculated $C_{12}$ as ref. [58],

$$C_{12} = C_{11} - 2C_{44}. \qquad (4)$$

Finally, Young's modulus and Poisson's ratio were determined as ref. [58],

$$E = \frac{(C_{11} - C_{12})(C_{11} + 2C_{12})}{C_{11} + C_{12}}, \qquad (5)$$

$$v = \frac{C_{12}}{C_{11} + C_{12}}. \qquad (6)$$

In addition, ultimate strength tests were performed in a similar manner to the above estimation of the elastic modulus, but using stepwise elongations of $\varepsilon = 0.01$.

**MD simulations of fracture toughness.** To compute the fracture toughness by MD simulations, we employed the method introduced by Brochard et al.[36] and subsequently applied and validated in various studies[37,38,59] (these references also provide the detailed introduction to the method). Initially the quenched structures were replicated into $2 \times 6 \times 4$ supercells (thus $1 \times 3 \times 2$ replications of the quenched structures) of 14208 atoms. In this structure, the atoms were removed in an ellipsoidal cylinder of length 36 Å and height 4 Å in the center of the simulation box to induce a precrack. This removal leaves a number of organic linkers incomplete. To ensure proper stability of the following simulation, we removed all incomplete linkers by identifying C, H, and N atoms of inadequate coordination numbers (using cutoffs of 2 Å for C–C and C–N bonds, and 1.5 Å for C–H bonds). Before fracture toughness simulations, the precracked structures were equilibrated in the $NPT$ ensemble at 300 K for 100 ps at zero pressure, which was found to be a reasonable time to obtain convergence of potential energy of the new structure. This equilibration was followed by a potential energy minimization. The ensemble was now changed to $NVT$ and was then equilibrated for another 10 ps. This was followed by step wise elongation of the simulation box of 1% every 5 ps. This step rate is more than an order of magnitude slower than what has previously been found to cause rate dependency of the fracture characteristics of other systems[59,60]. After each step, the structure was allowed to relax for 2.5 ps, followed by averaging of the pressure in the direction of elongation ($z$) for another 2.5 ps. The recorded pressure was used to plot a stress–strain curve up until and including fracture, which subsequently was integrated to yield the critical energy release rate under the assumption of elongation in the $z$-direction,

$$G_C = \frac{L_x L_y}{\Delta A_\infty} \int_{L_{z,0}}^{L_{z,max}} \sigma_z dL_z, \qquad (7)$$

where $L$ is the length of the specified direction, $\Delta A_\infty$ is created crack surface area during fracture[61], and $\sigma_z$ is the recorded pressure in the $z$-direction. By assuming $\gamma = G_C/2$, $K_{Ic}$ was finally estimated by the Irwin formula,

$$K_{Ic} = \sqrt{\frac{2\gamma E}{1 - v^2}} \qquad (8)$$

using the Young's modulus and Poisson's ratio from the simulations.

**MD simulations of surface energy.** To evaluate the brittleness parameter $(B_{index})$, we used the quenched $2 \times 2 \times 2$ supercells, which were initially equilibrated for 10 ps in the $NVT$ ensemble. The bulk potential energy was then estimated from this equilibration period. Two largely planar surfaces were then induced by splitting

the simulation box in two, yet only allowing Zn–N bonds to break. Afterwards the structure was equilibrated for another 50 ps, which was found to give a reasonable potential energy convergence. The difference between the potential energy before and after fracture process gave a measure for the surface energy ($\gamma_s$) used in the calculation of $B_{index}$. In the calculation of $\gamma_s$, we estimated the actual induced surface area, but found only minor deviations from planarity. We note that $\gamma_s$ is different from $\gamma$ in that $\gamma_s$ is exclusively taking surface energies into account, while $\gamma$ incorporates effects of plastic deformation and various rearrangements occurring during fracture.

**MD simulations of atomic stress**. We evaluated atomic stress using the incorporated stress/atom function in LAMMPS[56]. Note that, although stress is intrinsically a macroscopic property that is not properly defined for individual atoms, the "stress per atom" simply captures the contribution of each atom to the virial of the system and, hence, can be used to assess which elements are carrying the macroscopic stress imposed on the bulk glass. We followed the general scheme as for the estimation of $K_{Ic}$. To remove the effect of atomic velocity on stress, we performed a potential energy minimization before forcing atoms at rest only while performing the stress calculation. The output stress of LAMMPS is in a stress × volume formulation; hence, to obtain the presented stress results, we divide by the atomic volume of each elemental species as computed by the average of Voronoi volumes of each element (from bulk structures).

## Data availability

The data supporting the results within this paper are available from the corresponding author upon reasonable request. The source data underlying Figs. 1a, b, 1d, 2c, d, 3c, d, Supplementary Figs. 1, 3, 4, 5, 6, 7a-d, 8, Table 1 and Supplementary Table 1 are provided as a Source Data File.

## Code availability

The code used for the molecular dynamics simulations within this paper is available from the corresponding author upon reasonable request.

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

## Acknowledgements

The authors acknowledge K. Januchta and M.B. Østergaard for valuable discussions, and T. Kirwa for help with producing the three-point bending setup. This work was supported by VILLUM FONDEN under research grant no. 13253. M.B. acknowledges funding from the National Science Foundation under grant no. 1762292. We also thank Aalborg University for funding access to computational resources at the DeIC National HPC Center (ABACUS 2.0) at University of Southern Denmark and CLAAUDIA at Aalborg University.

## Author contributions

M.M.S. and Y.Y. conceived the study. T.T., S.S.S., and M.M.S. planned the experiments and simulations. M.S., A.Q., and T.T. prepared the samples. T.T. and L.R.J. performed the mechanical testing. S.S.S. performed the MD simulations and analyses, with input from M.B. All authors participated in discussing the data. T.T., S.S.S., and M.M.S. wrote the manuscript with revisions from Y.Y. and M.B.

## Competing interests

The authors declare no competing interests.
