## [Peer Review File · Nature Communications]

Reviewers' comments:

Reviewer #1 (Remarks to the Author):

The authors performed a combined experimental and atomic simulation studies on strength and fracture mechanics of Metal organic frameworks. The authors compared the correlation functions from experiments and MD simulations, the results are very comparable (The location of the peaks) to first validate their MD simulations. I give them a high credit for that. The experimental fracture toughness was measured via 3 point bending of pre-notched beams and compared to MD fracture toughness estimated from force-displacement curves of a sample with an initial slot. They studied the crack propagation mechanisms at the atomic scale via MD simulations, and they reported fracture toughness values that are very consistent between the experiment and models. Overall, the paper has presented new findings on fracture properties of MOFs with both experiments and MD simulations. However, the novelty of the work is not significant as mechanics of MOFs have been studied a lot. Also, technological significance of the work does not seem to be broad. I think this paper will be better suited for a specialty journal. My further comments are below.

- A search in the literature points to several studies on mechanical properties of MOFs. In what way is the experiments of the PIs adding to the literature considerably? Example is a critical review in Chem. Soc. Rev., 2011, 40, 1059-1080. A much more thorough lit review is required.
- What is the significance of the work? What applications of MOFs directly deals with mechanical performance? What is the larger community that can benefit from their findings?
- The difference in the strength between the experiment and model (experimental strength is 2 orders of magnitude less than MD simulations) could be attributed to the very different sample sizes used in the two and the much higher likelihood of finding defects with critical dimensions in the experimental sample. This explanation takes into account the very low fracture toughness of the material which makes it susceptible to initial flaws. Note that this analysis does not require the MD and experiment to have different flaw size distribution (per unit volume). The much smaller size of MD lowers the likelihood of having larger defects. The interpretation of authors in attributing the discrepancy to surface flaws does not seem sufficient.
- The discussion of the authors in demonstration of the significance of the coordinative bonds on fracture toughness is not straightforward and very hard to follow. Major rewording is required.
- How are pre-cracks made for the fracture experiment in such a brittle material?

Reviewer #2 (Remarks to the Author):

Metal-organic framework (MOF) glasses, as a new type of glass pioneered recently by the authors in current manuscript, demonstrate great potential applications due to the unique thermal, structural, and chemical properties. Unfortunately, the key mechanical properties, especially toughness and strength, of this kind of MOF glasses are still unknown due to the challenge in preparing large bulk glass samples for testing. More recently, the authors in current manuscript further developed a new MOF glass—zeolitic imidazolate framework-62 (ZIF-62) possessing the ultrahigh glass-forming ability superior to other glass materials. What's more, this kind of ZIF-62 glass can be made into large blocks so that the toughness and strength tests are possible based on the self-consistent single-edge precracked beam method. This paper addresses this problem, and firstly studies the fracture behavior of ZIF-62 glass in terms of its fracture toughness and flexural strength. To understand the microscopic mechanism, the current authors have done a lot of valuable theoretical simulations, and the theoretical and experimental results are in good agreement. This work is very thorough and detailed, and the current discovery is of great value for the further practical application of this kind of MOF glass materials. I'm glad that such important and excellent work can be published in Nature Communications. Before publishing, I only have two minor comments.

1)As is well known, the theoretical simulation of the glass structure is very challenging. Obviously, the

current author has made great efforts to complete this challenge. The author should provide the amorphous structure information in supporting information, such as CIF file, so that it's convenient for others to repeat these calculations.

2) It is unusually and interestingly discovered that the ZIF-62 glass shows a high Poisson's ratio but much lower fracture surface energy. To see if other MOF glasses have this relationship, it is better to add one or two examples in Fig. 3d. In addition, are the Poisson's ratios of MOF glasses all in the range of 0.3-0.4?

Reviewer #3 (Remarks to the Author):

Fracture Toughness of a Metal-Organic Framework Glass

Glasses based on metal organic frameworks are emerging new glassy materials with interesting properties. They are at the cutting-edge of inorganic chemistry and materials sciences, as well as the glassy physics. Usually, these MOF-glasses are difficult to prepare, this limits many fundamental studies like mechanical properties and deformation mechanisms. The data of mechanical toughness are thus very scarce in literatures.

In this work, the authors combined experiment and simulations to reveal the toughness(KIC), mechanical behaviors and the deformation mechanism of a ZIF62 glass in a comprehensive way. The KIC is obtained to be $0.1 \text{ MPa}\cdot\sqrt{\text{m}}$, which is even lower than that of brittle oxide glasses. As far as I know, this might be the first the report of KIC for MOF glasses. The very brittle nature is revealed by MD simulations to be the resultant of weak Zn-N coordinative bonds. A comparison of the toughness vs Poisson's ratio show the ZIF62 glasses are the ductile-to-brittle transition region in the correlation map for different glasses.

The paper is well prepared, the figures are OK and the discussions are clear and insightful. In sum, I would recommend this work to be published as is.

Response Letter

Reviewer #1 (Remarks to the Author):

The authors performed a combined experimental and atomic simulation studies on strength and fracture mechanics of Metal organic frameworks. The authors compared the correlation functions from experiments and MD simulations, the results are very comparable (The location of the peaks) to first validate their MD simulations. I give them a high credit for that. The experimental fracture toughness was measured via 3 point bending of pre-notched beams and compared to MD fracture toughness estimated from force-displacement curves of a sample with an initial slot. They studied the crack propagation mechanisms at the atomic scale via MD simulations, and they reported fracture toughness values that are very consistent between the experiment and models. Overall, the paper has presented new findings on fracture properties of MOFs with both experiments and MD simulations. However, the novelty of the work is not significant as mechanics of MOFs have been studied a lot. Also, technological significance of the work does not seem to be broad. I think this paper will be better suited for a specialty journal. My further comments are below.

Response: We thank the reviewer for acknowledging the new findings and their validity. Although we agree that the mechanical properties of various crystalline MOFs have been thoroughly studied, this is not yet the case for glassy MOFs (see also below). To the best of our knowledge, the present study is the first one to explore the fracture toughness of **melt-quenched bulk** glasses. This exploration is critically important for assessing the mechanical performances of MOF bulk glasses in order to find their suitable application fields. Furthermore, MOF glasses are a strong case to understand the bonding and structural effect on the fracture behavior of melt-quenched glasses, and thus the present work is of general interest to disordered materials.

Indeed, knowledge about the mechanical properties of crystalline MOFs are likely to be insufficient to understand the mechanical properties of glassy MOFs, especially in terms of fracture mechanics. For example, a clear dependence of crystal direction on fracture behavior and fracture toughness of some hybrid frameworks has been reported (Tan et al. *Acta Mater.*, 2009, **57**, 3481), but such dependence is not expected in isotropic glasses. More generally, non-crystalline materials such as glasses lack the defects that provide toughening and flaw insensitivity in many crystalline solids. We have clarified these differences and the novelty of the present study in the revised manuscript (pp. 2-3).

- A search in the literature points to several studies on mechanical properties of MOFs. In what way is the experiments of the PIs adding to the literature considerably? Example is a critical review in *Chem. Soc. Rev.*, 2011, 40, 1059-1080. A much more thorough lit review is required.

Response: As noted above, it is important to distinguish between studies on crystalline and glassy MOFs. MOF glasses were discovered in 2015 and, to our knowledge, there are only two studies on MOF glass fracture mechanics (refs. 11 and 15 in the main text). Both these studies use indentation methods and, hence, are intrinsically unable to offer any information on the strength and fracture toughness of MOF glasses. As such, this is the first report on the mechanisms controlling the fracture toughness of a glassy MOF—and, from a practical viewpoint, this is simply the first time that the value of the fracture toughness of a glassy MOF is ever reported. More generally, this is also the first report on the fracture toughness of a MOF (glassy or crystalline) measured by a non-indentation method.

However, we do agree with the reviewer that it would be helpful to provide a literature review on the studies of mechanical properties on crystalline MOF. This has been done on p. 2 of the revised manuscript. The suggested reference has been cited, which is an excellent overview about the mechanical behavior of crystalline MOFs and is helpful for understanding their counterpart - glasses.

- What is the significance of the work? What applications of MOFs directly deals with mechanical performance? What is the larger community that can benefit from their findings?

Response: Our manuscript deals with the fundamental understanding of fracture mechanics of a representative MOF glass. We show that the origin of its low fracture toughness is due to the presence of the weak coordinative Zn-N bonds, in turn giving rise to an anomalous brittle-to-ductile transition behavior (Fig. 3d). These findings have implications for the future design of MOF glasses with tailored mechanical properties. However, considering that MOF glasses were only discovered 5 years ago, there are currently no industrial applications. A number of possible applications of MOF glasses have been suggested in the literature (see, e.g., Tao et. al., *Adv. Mater.*, 2017, **29**, 1601705). These points have been stated on p. 3 of the revised manuscript.

- The difference in the strength between the experiment and model (experimental strength is 2 orders of magnitude less than MD simulations) could be attributed to the very different sample sizes used in the two and the much higher likelihood of finding defects with critical dimensions in the experimental sample. This explanation takes into account the very low fracture toughness of the material which makes it susceptible to initial flaws. Note that this analysis does not require the MD and experiment to have different flaw size distribution (per unit volume). The much smaller size of MD lowers the likelihood of having larger defects. The interpretation of authors in attributing the discrepancy to surface flaws does not seem sufficient.

Response: Good point. We agree with the reviewer that the probed system size in the simulations puts a limitation on the obtainable defect size and thus lowers the probability of finding critical flaws in the simulated glass. However, we maintain the argument that the lack of surface in the simulated glass specimen also contributes to the observed discrepancy in strength between experiments and simulations. We have revised the manuscript to discuss both possibilities (p. 10).

- The discussion of the authors in demonstration of the significance of the coordinative bonds on fracture toughness is not straightforward and very hard to follow. Major rewording is required.

Response: We have reworded and rephrased the related sections in the manuscript on pp. 11-13.

- How are pre-cracks made for the fracture experiment in such a brittle material?

Response: This is a good question. Considering the low fracture toughness ($K_{Ic} \sim 0.1 \text{ MPa m}^{0.5}$) of the studied glass, it was very challenging to prepared samples for the single edge precrack beam (SEPB) method used to determine K_{Ic} . When performing cutting and polishing, and inducing the precrack, there is great risk of breaking the sample, but we did succeed in producing the SEPB specimens with a good precrack. We acknowledge that more details than given in the initial submission are needed to fully replicate the performed experiments. This has now been done in the revised Methods section (p. 17). Specifically, we have explained how the utilized compression fixture puts the lower part (indented or groove part) of the specimen in tension, while the upper part is in compression. The tensile stress opens the cracks from the five indents, enabling the crack to grow. However, it stops when it reaches the compressive stress, thus creating the desired pre-crack.

Reviewer #2 (Remarks to the Author):

Metal-organic framework (MOF) glasses, as a new type of glass pioneered recently by the authors in current manuscript, demonstrate great potential applications due to the unique thermal, structural, and chemical properties. Unfortunately, the key mechanical properties, especially toughness and strength, of this kind of MOF glasses are still unknown due to the challenge in preparing large bulk glass samples for testing. More recently, the authors in current manuscript further developed a new MOF glass—zeolitic imidazolate framework-62 (ZIF-62) possessing the ultrahigh glass-forming ability superior to other glass materials. What's more, this kind of ZIF-62 glass can be made into large blocks so that the toughness and strength tests

are possible based on the self-consistent single-edge precracked beam method. This paper addresses this problem, and firstly studies the fracture behavior of ZIF-62 glass in terms of its fracture toughness and flexural strength. To understand the microscopic mechanism, the current authors have done a lot of valuable theoretical simulations, and the theoretical and experimental results are in good agreement. This work is very thorough and detailed, and the current discovery is of great value for the further practical application of this kind of MOF glass materials. I'm glad that such important and excellent work can be published in Nature Communications. Before publishing, I only have two minor comments.

Response: We thank the reviewer for recognizing the novelty and importance of the work.

1)As is well known, the theoretical simulation of the glass structure is very challenging. Obviously, the current author has made great efforts to complete this challenge. The author should provide the amorphous structure information in supporting information, such as CIF file, so that it's convenient for others to repeat these calculations.

Response: An example structure of one of the eight amorphous structures will be provided as a supporting file (in LAMMPS data file format) along with the revised manuscript. This has been noted in the Methods section of the revised manuscript (p. 18), where we have also provided a reference for details about the data format and software for possible formatting.

2)It is unusually and interestingly discovered that the ZIF-62 glass shows a high Poisson's ratio but much lower fracture surface energy. To see if other MOF glasses have this relationship, it is better to add one or two examples in Fig. 3d. In addition, are the Poisson's ratios of MOF glasses all in the range of 0.3-0.4?

Response: We agree with the reviewer that adding such data to Fig. 3d would be very interesting. Unfortunately, this is currently not possible to our knowledge. Bulk samples of sufficient size for fracture energy measurement (using a non-indentation method) can currently not be made for MOF glasses, besides ZIF-62 glass.

Reviewer #3 (Remarks to the Author):

Fracture Toughness of a Metal-Organic Framework Glass

Glasses based on metal organic frameworks are emerging new glassy materials with interesting properties. They are at the cutting-edge of inorganic chemistry and materials sciences, as well as the glassy physics. Usually, these MOF-glasses are difficult to prepare, this limits many fundamental studies like mechanical properties and deformation mechanisms. The data of mechanical toughness are thus very scarce in literatures.

In this work, the authors combined experiment and simulations to reveal the toughness(KIC), mechanical behaviors and the deformation mechanism of a ZIF62 glass in a comprehensive way. The KIC is obtained to be $0.1 \text{ MPa}\cdot\sqrt{\text{m}}$, which is even lower than that of brittle oxide glasses. As far as I know, this might be the first the report of KIC for MOF glasses. The very brittle nature is revealed by MD simulations to be the resultant of weak Zn-N coordinative bonds. A comparison of the toughness vs Poisson's ratio show the ZIF62 glasses are the ductile-to-brittle transition region in the correlation map for different glasses.

The paper is well prepared, the figures are OK and the discussions are clear and insightful. In sum, I would recommend this work to be published as is.

Response: We thank the reviewer for the kind comments about our work.

REVIEWERS' COMMENTS:

Reviewer #1 (Remarks to the Author):

My comments to author's responses are below:

About the first bullet point, the authors comment that their work is novel as it deals with glassy MOFs is well taken. However, I'm afraid this will considerably reduce the broad appeal of the work, and as such makes it more suitable for a specialty journal.

Although the authors maintain that they are doing fundamental research, the value of their fundamental research should be evaluated in terms of the potential benefits of the material for the society. In this regard, I strongly urge the authors to add a few more works explaining the new opportunities that knowing the mechanical properties of MOFs can open for the society. Other than that, the paper is good to go.

Reviewer #2 (Remarks to the Author):

I have carefully read all the replies from the authors to the three reviewers. I think the authors have replied to the reviewers' comments with satisfaction and made their best modifications. So I recommend publishing this excellent work in Nature Communications.

Response Letter

Reviewer #1 (Remarks to the Author):

My comments to author's responses are below:

About the first bullet point, the authors comment that their work is novel as it deals with glassy MOFs is well taken. However, I'm afraid this will considerably reduce the broad appeal of the work, and as such makes it more suitable for a specialty journal.

Response: We of course acknowledge that the field of MOF glasses is smaller than that of MOF crystals. This is natural since the first MOF glasses were discovered in 2015, but the field has been receiving increasing attention since that time, with numerous publications in broad appeal journals, even in Science (Madsen et al. Science 2020). In addition, understanding of the mechanical properties of MOF glasses is an important addition to the big picture of the mechanical properties of different families of glass materials, e.g., regarding the impact of the type of chemical bonds and structure on fracture behavior. In this regard, we believe the present work is of interest for a broad readership.

Although the authors maintain that they are doing fundamental research, the value of their fundamental research should be evaluated in terms of the potential benefits of the material for the society. In this regard, I strongly urge the authors to add a few more works explaining the new opportunities that knowing the mechanical properties of MOFs can open for the society. Other than that, the paper is good to go.

Response: Thank you for the suggestion. Since MOF glasses are still young, the application fields are currently being extended. Following the reviewer's suggestion, we have expanded our discussion of this on p. 3 to highlight some of the possible application fields (and cited relevant references), such as gas storage, membranes, and photonics. To make structurally and mechanically stable glasses for such applications, knowledge of the mechanical properties is required.

Reviewer #2 (Remarks to the Author):

I have carefully read all the replies from the authors to the three reviewers. I think the authors have replied to the reviewers' comments with satisfaction and made their best modifications. So I recommend publishing this excellent work in Nature Communications.

Response: Thank you for the kind comments.